# Hypermethylation of DNA Methylation Markers in Non-Cirrhotic Hepatocellular Carcinoma

**DOI:** 10.3390/cancers15194784

**Published:** 2023-09-28

**Authors:** Siyu Fu, Teoman Deger, Ruben G. Boers, Joachim B. Boers, Michael Doukas, Joost Gribnau, Saskia M. Wilting, José D. Debes, Andre Boonstra

**Affiliations:** 1Department of Gastroenterology and Hepatology, Erasmus MC, University Medical Center, 3015 CN Rotterdam, The Netherlands; s.fu@erasmusmc.nl (S.F.); debes003@umn.edu (J.D.D.); 2Department of Medical Oncology, Erasmus MC Cancer Institute, University Medical Center, 3015 GD Rotterdam, The Netherlandss.wilting@erasmusmc.nl (S.M.W.); 3Department of Developmental Biology, Erasmus MC, University Medical Center, 3015 GD Rotterdam, The Netherlands; r.g.boers@erasmusmc.nl (R.G.B.); j.gribnau@erasmusmc.nl (J.G.); 4Department of Pathology, Erasmus MC, University Medical Center, 3000 CA Rotterdam, The Netherlands; 5Department of Medicine, University of Minnesota, Minneapolis, MN 55455, USA

**Keywords:** non-cirrhotic HCC, DNA methylation markers, qMSP, MeD-seq, HOXA1, CLEC11A, AK055957, TSPYL5

## Abstract

**Simple Summary:**

Hepatocellular carcinoma (HCC) remains a significant cause of cancer-related deaths worldwide. Although most HCC cases have a background of cirrhosis, up to 20–30% of patients develop HCC in a non-cirrhotic liver. The prognosis of these unusual HCC cases is poor, since surveillance is not common. Therefore, investigating sensitive and specific biomarkers to detect non-cirrhotic HCC is crucial. Aberrant DNA methylation has been reported to play an important role in the development of cirrhotic HCC, while limited information can be found on non-cirrhotic HCC. This study is the first to determine the performance of reported promising DNA methylation markers in non-cirrhotic HCC. A total of 146 liver tissues were tested for 4 methylation markers using PCR- and sequencing-related techniques. We demonstrated significant DNA methylation changes in non-cirrhotic HCC compared to non-HCC benign lesions. These findings may be highly relevant to the future application of DNA methylation markers in non-cirrhotic HCC surveillance.

**Abstract:**

Aberrant DNA methylation changes have been reported to be associated with carcinogenesis in cirrhotic HCC, but DNA methylation patterns for these non-cirrhotic HCC cases were not examined. Therefore, we sought to investigate DNA methylation changes on non-cirrhotic HCC using reported promising DNA methylation markers (DMMs), including HOXA1, CLEC11A, AK055957, and TSPYL5, on 146 liver tissues using quantitative methylation-specific PCR and methylated DNA sequencing. We observed a high frequency of aberrant methylation changes in the four DMMs through both techniques in non-cirrhotic HCC compared to cirrhosis, hepatitis, and benign lesions (*p* < 0.05), suggesting that hypermethylation of these DMMs is specific to non-cirrhotic HCC development. Also, the combination of the four DMMs exhibited 78% sensitivity at 80% specificity with an AUC of 0.85 in discriminating non-cirrhotic HCC from hepatitis and benign lesions. In addition, HOXA1 showed a higher aberrant methylation percentage in non-cirrhotic HCC compared to cirrhotic HCC (43.3% versus 13.3%, *p* = 0.039), which was confirmed using multivariate linear regression (*p* < 0.05). In summary, we identified aberrant hypermethylation changes in HOXA1, CLEC11A, AK055957, and TSPYL5 in non-cirrhotic HCC tissues compared to cirrhosis, hepatitis, and benign lesions, providing information that could be used as potentially detectable biomarkers for these unusual HCC cases in clinical practice.

## 1. Introduction

Liver cancer is a major cause of cancer-related deaths worldwide [1]. Hepatocellular carcinoma (HCC) originates from hepatocytes and accounts for more than 90% of primary liver cancers [2]. Most patients with HCC have a background of cirrhosis, but up to 20–30% of HCC cases can develop in a non-cirrhotic liver [3]. The main risk factors for HCC differ geographically. Hepatitis B virus (HBV) and hepatitis C virus (HCV) infection are the major causes of HCC in Asia and most African regions. However, non-viral causes, such as alcoholic liver disease (ALD) and non-alcoholic fatty liver disease (NAFLD), significantly contribute to the number of HCC cases in North America, Europe, Latin America, and Australasia [4]. The overall 5-year survival rate of patients with HCC in the United States is only 19.6%, but can be as low as 2.5% for late-stage HCC [5]. One of the reasons for this high mortality is that patients often do not present any symptoms until the disease has reached an advanced stage [6]. Hence, early-stage detection of HCC is extremely important, since a curative treatment is not feasible for patients diagnosed at late stages, thereby underscoring the need for sensitive and specific biomarkers that can identify early-stage HCC in high-risk populations [7].

Guidelines from the American Association for the Study of Liver Diseases (AASLD) recommended that individuals with cirrhosis undergo HCC surveillance through ultrasonography with or without alpha-fetoprotein (AFP) every 6 months [8]. However, the sensitivity of ultrasonography in detecting early-stage HCC is only 47%, and when combined with AFP, this increases to just 63% [9]. Efforts to explore novel biomarkers for early-stage HCC detection, such as AFP-L3 [10], des-gamma-carboxy prothrombin (DCP) [10], and DNA methylation markers (DMMs), are ongoing [11,12].

Aberrant DNA methylation is one of the epigenetic mechanisms involved in human cancer development, including HCC [13]. It may lead to the inactivation of tumor suppressor genes or the activation of cancer-related genes promoting carcinogenesis [14]. Generally, cancer-related DMMs are identified using DNA methylation-related sequencing in a small number of diseased liver tissues, followed by validation in tissues or cell-free DNA (cfDNA) from plasma by PCR-related techniques [13]. Based on recent studies, and as recently reviewed by us [15], DNA hypermethylation of *HOXA1*, *CLEC11A*, *AK055957*, and *TSPYL5* was observed in HCC as compared to non-HCC, both in genomic DNA from liver tissues and cfDNA from plasma [13,15]. In blood, combining DMMs (such as *HOXA1* and *TSPYL5*) with protein biomarkers (AFP, AFP-L3) or demographic factors exhibited 71–82% sensitivity for the early-stage HCC detection in two large clinical cohorts, which was superior over AFP and the combination of AFP, AFP-L3, and DCP [16,17]. DNA hypermethylation of *HOXA1* and *TSPYL5* has been validated in three large clinical cohorts, and *CLEC11A* and *AK055957* exhibited hypermethylation levels in HCC tissues compared to cirrhotic controls in a methylation intensity map [13], while limited information about the four DMMs in non-cirrhotic HCC can be found. Therefore, investigating the methylation levels of the four DMMs in non-cirrhotic HCC might be helpful for non-cirrhotic HCC detection.

The goal of this study is to conduct a comprehensive evaluation of the DNA methylation levels of the recently described set of promising DMMs, *HOXA1*, *CLEC11A*, *AK055957*, and *TSPYL5*, in liver tissues from non-cirrhotic HCC, and to compare them to cirrhotic HCC, cirrhotic livers, non-cirrhotic livers, and benign lesions (adenoma and focal nodular hyperplasia). In addition, we took advantage of some HCC samples and controls for validation using the recently developed method of genome-wide methylation DNA sequencing of LpnPI-digested fragments (MeD-seq), which covers more than 50% of the cytosine-guanine (CpG) dinucleotides without the need for harsh bisulfite conversion. Through a detailed analysis of the association between cirrhosis status, liver etiology, gender, age, and tumor size with DNA methylation changes, we identified DNA methylation changes in *HOXA1-*, *CLEC11A-*, *AK055957-*, and *TSPYL5*-associated clinical risk factors.

## 2. Materials and Methods

### 2.1. Study Population and Samples

The study evaluates DNA methylation marker levels in the genomic DNA of snap-frozen liver tissues from archived samples from the Erasmus Medical Center. HCC liver tissues were sampled during segmental surgical resection or were biopsied from patients before regional therapy or systemic chemotherapy. All samples had previously been subjected to pathological diagnosis, and HBV or HCV infections were diagnosed serologically. Patients with sufficient information regarding their cirrhosis status and liver disease etiology, including viral, non-viral, and cryptogenic etiology, were included. Patients were excluded in cases of mixed-etiology HCC, non-HCC liver malignancy, and HCC co-existing with other malignancies. A total of 146 individuals were included in the quantitative methylation-specific PCR (qMSP) analysis, including non-cirrhotic HCC (n = 60), cirrhotic HCC (n = 15), cirrhotic non-HCC livers (n = 30), hepatitis (n = 34), and benign lesions from independent patients (n = 7; 4 adenoma, 3 focal nodular hyperplasia) (Appendix A). We included 37 ALD-related liver disease cases, including 8 non-cirrhotic HCC, 4 cirrhotic HCC, 10 hepatitis, and 15 cirrhosis, as well as 7 benign lesions, in the MeD-seq analysis. Of the HCC samples, 64 of 75 (85.3%) with sufficient liver tissues were evaluated by hematoxylin and eosin (H&E) staining to confirm the type of tissue (tumorous or non-tumorous) and the pathology. Patients with HBV or HCV infection were assigned to the viral-related group, patients with ALD-related liver disease or NAFLD-related liver disease to the non-viral group, and patients in whom all other etiologies were excluded in the cryptogenic group. The serum AFP levels were measured as part of routine diagnostics.

### 2.2. DNA Isolation and Quantification

Genomic DNA was extracted from frozen liver tissues using a QIAamp DNA Mini Kit (Qiagen, Venlo, The Netherlands). Briefly, small pieces of tissues were cut and transferred to 1.5 mL Eppendorf tubes. Afterward, all samples were incubated at 56 °C in DNA extraction buffer containing proteinase K for 3–5 h, according to manufacturer’s guidelines. After complete digestion, DNA was eluted by AE buffer (10 mM Tris·Cl; 0.5 mM EDTA; pH 9.0), and the concentration was quantified with NanoDrop (ThermoFisher, Wilmington, DE, USA). DNA samples were stored at −80 °C for the DNA methylation analysis.

### 2.3. Quantitative Methylation-Specific PCR (qMSP)

A total of 750 ng of genomic DNA was bisulfite-converted and eluted using 15 µL of elution buffer via the EZ DNA Methylation kit (Zymo Research, Irvine, CA, USA), of which 2 µL was used for each DNA methylation analysis using the EpiTect MethyLight Master Mix (Qiagen). The primers were designed to amplify the methylated DNA sequence, and the resulting amplicons were quantified using TaqMan probes (Appendix A). Primers and probes were ordered from Eurogentec (Seraing, Belgium). The specificity of the primers was checked using the EpiTect PCR Control DNA Set (Qiagen), which contains bisulfite-converted methylated and unmethylated DNA, as well as unconverted unmethylated DNA. All the primers and probes used in the assay were positive for bisulfite-converted methylated DNA, while negative for bisulfite unmethylated DNA and unconverted unmethylated DNA. The DNA methylation levels of *HOXA1*, *CLEC11A*, *AK055957*, and *TSPYL5* were measured in single qMSP assays, and the modified, unmethylated sequence of the housekeeping gene β-actin (*ACTB*) was amplified as a reference [18]. qMSP reactions were carried out in a 12.5 µL reaction volume containing 2 µL bisulfite-converted genomic DNA, 400 nM per primer, 200 nM probe, 6.25 µL 2 × EpiTect MethyLight Master Mix (w/o ROX) using the StepOnePlus™ Real-Time PCR System (ThermoFisher). Only samples with a Ct ≤ 32 for *ACTB* were considered to have sufficient DNA and adequate bisulfite conversion amounts, and these were selected for data analysis, resulting in 146 samples being included in this analysis. The DNA methylation levels were normalized to *ACTB* using the comparative Ct method (2 − ΔCT) [19].

### 2.4. Methylated DNA Sequencing (MeD-Seq)

MeD-seq assays were performed as previously described [20]. Briefly, 20 µL genomic DNA (input 90 ng) from frozen liver tissues was digested with LpnPI (New England Biolabs), generating 32 bp fragments around the methylated recognition site containing a CpG. These short DNA fragments were further processed using the ThruPlex DNA–seq 96D kit (Rubicon Genomics, Ann Arbor, MI, USA). Stem–loop adapters were blunt-end ligated to repaired input DNA, then amplified to include dual-indexed barcodes using a high-fidelity polymerase to generate an indexed Illumina NGS library. The amplified end product was purified on a Pippin HT system with 3% agarose gel cassettes (Sage Science, Beverly, MA, USA). Multiplexed samples were sequenced on Illumina NextSeq2000 systems for paired-end reads of 50 bp, according to the manufacturer’s instructions. The dual-indexed samples were de-multiplexed using bcl2fastq v2.20 software (Illumina, San Diego, CA, USA).

### 2.5. MeD-Seq Data Analysis

Data processing was carried out using specifically created scripts in Python. Raw fastq files were subjected to Illumina adaptor trimming and reads were filtered based on LpnPI restriction site occurrence between 13–17 bp from either the 5′ or 3′ end of the read. Reads that passed the filter were mapped to hg38 using bowtie2. Genome-wide individual LpnPI site scores were used to generate read count scores for the following annotated regions: transcription start sites (TSS, 1 kb before and 1 kb after), CpG-islands and gene bodies (1 kb after TSS till TES). Gene and CpG-island annotations were downloaded from ENSEMBL (Homo_sapiens_hg38.GRCh38.79.gtf, www.ensembl.org, accessed on 23 March 2023. In addition, a genome-wide sliding window was used to detect sequentially differentially methylated LpnPI sites. Statistical significance was assessed between LpnPI sites in predetermined groups using the chi-square test. Neighboring significant LpnPI sites were binned and reported. Our annotation of the overlap of genome-wide regions detected differentially methylated regions (DMRs) reported for TSS, CpG-islands, and gene body regions. The DMR thresholds were based on the LpnPI site count, DMR sizes (in bp), and fold changes of the read counts, as mentioned in the figure legends, before hierarchical clustering was performed.

### 2.6. Statistics

Statistical analyses were performed using SPSS 28.0.1.0 (SPSS Inc., Chicago, IL, USA). Continuous variables are presented as medians, and categorical variables as percentages. The Mann–Whitney U test was used for testing continuous variables, and chi-square χ^2^ tests or Fisher’s exact test for dichotomous variables were employed when appropriate. The areas under the curves (AUCs) were also used to assess the performance of biomarkers in terms of discriminating HCC from non-HCC. Multivariate linear regression analysis was performed to test whether any clinical risk factors were associated with the DNA methylation levels. A two-tailed value of *p* < 0.05 was considered statistically significant.

## 3. Results

### 3.1. Hypermethylation of AK055975 and TSPYL5 Was Observed in Cirrhotic HCC When Compared to Cirrhotic Livers

To evaluate the findings from a previously published set of 4 DMMs that were all hypermethylated in cirrhotic HCC compared to cirrhosis [13] in our cohort, we first assessed, by means of qMSP, the methylation changes of *HOXA1*, *CLEC11A*, *AK055957*, and *TSPYL5* in the liver tissue of 15 cirrhotic HCC and 30 cirrhosis patients with mixed etiologies (for patient details see Appendix A). As shown in Figure 1A, *AK055957* and *TSPYL5* exhibited higher DNA methylation levels in cirrhotic HCC compared to cirrhosis (*p* < 0.05). In contrast, *HOXA1* and *CLEC11A* showed comparable DNA methylation levels between the two groups (*p* > 0.05). Although the individual biomarkers *HOXA1*, *CLEC11A*, and *AK055957* poorly discriminated between cirrhotic HCC and cirrhosis with 47–60% sensitivity (AUCs 0.52–0.72) (Figure 1B, Table 1), the combination of DMMs demonstrated higher AUCs compared to single biomarkers (AUCs 0.84–0.86). With a panel of four DMMs, the sensitivity increased to 87% at 80% specificity.

In addition, we observed considerable variation in the DNA methylation levels of the 4 DMMs within cirrhotic HCC tumors, which may be related to distinct clinical risk factors. We therefore analyzed the DNA methylation levels by assessing the underlying etiology of cirrhotic HCC compared to cirrhotic livers. A total of 17 viral-related liver diseases (8 HCC, 9 cirrhosis) and 20 non-viral-related liver diseases (5 HCC, 15 cirrhosis) were included in this analysis. As presented in Figure 1C, within the viral group, only *TSPYL5* showed significantly increased methylation levels in viral-related HCC compared to cirrhosis (*p* < 0.05); no differences were found for *HOXA1*, *CLEC11A*, or *AK055957*. In the non-viral group, albeit not significant, a trend was observed for higher methylation levels in HCC for *AK055957* and *TSPYL5*. These findings indicate that livers from cirrhotic HCC have different methylation patterns of *AK055957* and *TSPYL5* compared to cirrhotic non-HCC livers, and that etiology is an important factor that affects the DNA methylation levels.

### 3.2. Aberrant DNA Methylation Levels and Percentage of the Four DMMs in Non-Cirrhotic HCC When Compared to Non-Tumor Tissues Both in qMSP and MeD-Seq

Although, generally, HCC occurs in cirrhotic livers, up to 20% of HCC cases can develop into non-cirrhotic livers [21]. However, few studies have investigated DNA methylation levels in non-cirrhotic HCC tissues [13]. To explore this, we selected the liver tissues of non-cirrhotic HCC (n = 60) in order to compare them with those from patients with hepatitis (n = 34) and benign lesions (n = 7) as non-tumor control groups (Appendix A). As shown in Figure 2A, hypermethylation of *HOXA1*, *CLEC11A*, *AK055957*, and *TSPYL5* was observed in non-cirrhotic HCC compared to hepatitis and cirrhosis (*p* < 0.05). Except for *CLEC11A*, increased methylation levels of the other three DMMs were also observed in non-cirrhotic HCC compared to benign lesions (*p* < 0.05).

To validate the results of qMSP, we investigated the overall DNA methylation changes in patients with non-cirrhotic HCC using MeD-seq based on a balanced number of patients with ALD-related HCC and control groups. As shown in Appendix A, 37 liver samples from patients with ALD-related liver diseases were selected for MeD-seq analysis, including 8 non-cirrhotic HCC, 4 cirrhotic HCC, 15 cirrhotic, and 10 hepatitis samples. In addition, seven benign lesions were included. First, we restricted our analysis to only CpG islands and gene promotor regions. All, except *AK055957*, were part of the standard gene list from the UCSC platform (GRCh38/hg38, genome.ucsc.edu, accessed on 23 March 2023) and were included in the algorithm of MeD-seq. The regions of primer design from qMSP for *HOXA1*, *CLEC11A*, and *TSPYL5* were located in the regions of calculation by MeD-seq. As shown in Figure 2B, a comparison of the MeD-seq sequencing data demonstrated that the methylation levels of *HOXA1*, *CLEC11A*, and *TSPYL5* in non-cirrhotic HCC were higher than in the control groups: the hypermethylation scores of *HOXA1*, *CLEC11A*, and *TSPYL5* were able to discriminate non-cirrhotic HCC from cirrhosis, hepatitis, and benign lesions (*p* < 0.05), as is consistent with the qMSP data. In contrast, both MeD-seq and qMSP exhibited no distinct methylation changes in *HOXA1* during the comparison of cirrhotic HCC and cirrhosis. However, using MeD-seq, *TSPYL5* was unable to distinguish cirrhotic HCC from cirrhosis or hepatitis, while *CLEC11A* was, which differed from the qMSP data. Notably, the median methylation scores of *HOXA1*, *CLEC11A*, and *TSPYL5* were higher in cirrhotic HCC compared to cirrhosis, hepatitis, and benign lesions, although only *CLEC11A* exhibited a statistical difference (*p* < 0.05). Interestingly, the methylation scores of the three DMMs were relatively low in benign lesions, as is consistent with the previous qMSP data, indicating that the hypermethylation of these genes is low in benign lesions compared to malignant lesions. In summary, the data from MeD-seq confirmed the results of qMSP that non-cirrhotic HCC had the highest methylation levels of *HOXA1*, *CLEC11A*, and *TSPYL5* compared to hepatitis, benign lesions, and cirrhosis.

Based on the qMSP data, to discriminate cancer-related DNA methylation levels from non-tumors, we set the arbitrary threshold of aberrant DNA methylation levels to above the 95% percentile level in the hepatitis group, and used this threshold to determine the DNA methylation levels and percentages in non-cirrhotic HCC and cirrhotic HCC compared to the patients with cirrhosis, hepatitis, and benign lesions. Our evaluation of the DNA methylation percentage supported these findings (Figure 2C, Table 2); the hypermethylation percentage of the four DMMs was much higher in the non-cirrhotic HCC compared to the control groups. Also, in the livers of non-cirrhotic HCC, *HOXA1* exhibited a significantly higher percentage of hypermethylation than in cirrhotic HCC (43.3% versus 13.3%, *p* = 0.039, Table 2). Except for *HOXA1*, there were no differences in DNA methylation levels between non-cirrhotic HCC and cirrhotic HCC for the other three DMMs. In summary, we demonstrated that hypermethylation of the four DMMs is more specific in non-cirrhotic HCC than in hepatitis, benign lesions, and cirrhosis.

Next, we checked the performance of the four DMMs in non-cirrhotic HCC using the ROC curve (Figure 2D). The individual DMM only achieved 55–63% sensitivity at 80% specificity (AUCs 0.67–0.78) in discriminating non-cirrhotic HCC from hepatitis and benign lesions, while the combination of 4 DMMs significantly increased the sensitivity to 78% at 80% specificity (AUCs 0.84–0.85). In addition, we assessed the number of cases with serum AFP levels higher than 20 ng/µL and DMMs above the 95% percentile of hepatitis in non-cirrhotic HCC (Figure 2E). In the non-cirrhotic HCC group, AFP levels were available for 52 out of 60 patients. Of these, 31 had no elevated AFP levels (60%). The positivity levels for *HOXA1*, *CLEC11A*, *AK055957*, and *TSPYL5* in non-cirrhotic HCC were 40.4%, 53.8%, 50.0%, and 48.1%, respectively. Of the patients, 73.1% (38 out of 52) had at least one positive DMM in non-cirrhotic HCC, and when combined with AFP, this increased to 82.7% (43 out of 52). Therefore, DMMs and AFP might be complementary in non-cirrhotic HCC detection.

### 3.3. DNA Methylation Changes of the Four DMMs Were Not Associated with Clinical Factors Other Than Gender and Cirrhosis Status

Besides the cirrhosis status, it has been reported that gender, age, etiology, and tumor size also affect DNA methylation changes [22]. To assess this in our cohorts, we evaluated relevant clinical parameters with DNA methylation changes, including gender, age, etiology, tumor size, and cirrhosis status, using multivariate linear regression analysis (Table 3). We classified patients of different ages into two groups from young to old and two groups for different tumor sizes. As we found previously, non-cirrhotic HCC patients had much higher methylation levels of *HOXA1* compared to cirrhotic HCC patients (*p* < 0.05), which is consistent with the previously obtained methylation percentage data (Table 1). We also found that male HCC patients exhibited higher methylation levels of *TSPYL5* than females (*p* < 0.05). In contrast, although patients with tumor sizes over 5 cm showed higher methylation levels of *AK055957* (23.38 versus 3.00) than small tumors, it was not a significant risk factor contributing to the DNA methylation changes. Also, cryptogenic-related HCC exhibited enhanced the DNA methylation levels of *CLEC11A* and *AK055957* compared to viral- or non-viral-related HCC, but no statistical difference was detected among these groups. Similarly, in patients of different ages, the DNA methylation patterns were comparable (*p* > 0.05). The results demonstrate that gender and cirrhosis status are important clinical risk factors for the hypermethylation of *HOXA1* and *TSPYL5*, respectively. In contrast, no association with etiology, age, or tumor size could be demonstrated for the DNA methylation changes.

## 4. Discussion

In our study, through two different technologies, we demonstrated that non-cirrhotic HCC exhibited higher DNA methylation levels than cirrhotic livers and those affected by hepatitis and benign lesions. Currently, HCC in non-cirrhotic patients is often diagnosed at a late stage when patients become symptomatic, since surveillance is not feasible due to the absence of known risk factors [3]. Also, non-cirrhotic HCC represents a subgroup in which the mechanisms of hepatocarcinogenesis remain unclear. Importantly, and to further support these findings, we also found an increased percentage of DNA methylation in non-cirrhotic HCC compared to hepatitis, benign lesions, cirrhosis, and even cirrhotic HCC samples, such as *HOXA1*, which was validated by multivariate linear regression analysis. The results thus demonstrate that specific epigenetic changes can be observed and likely play a pronounced role in the development of non-cirrhotic HCC. In addition, the combination of DMMs significantly increased the sensitivity in terms of discriminating non-cirrhotic HCC from hepatitis and benign lesions, which might be applied to tackle the heterogeneity of HCC. More detailed studies on these DMMs may shed light on the mechanisms of carcinogenesis and may identify sensitive and specific biomarkers for detecting early-stage non-cirrhotic HCC in the future.

Importantly, we observed different DNA methylation patterns between non-cirrhotic HCC and benign lesions: the qMSP-based results exhibited higher methylation levels and percentages of *HOXA1*, *CLEC11A*, *AK055957*, and *TSPYL5* in non-cirrhotic HCC than benign lesions. Also, MeD-seq analysis demonstrated that the hypermethylation score of *HOXA1*, *CLEC11A*, and *TSPYL5* was higher in non-cirrhotic HCC than benign lesions, indicating that the transition from benign lesions to non-cirrhotic HCC involves significant DNA methylation changes. Therefore, the four DMMs could be applied as biomarkers for patients with benign lesions who are at a high risk of non-cirrhotic HCC. DNA methylation changes between non-cirrhotic HCC and benign lesions have not been reported before. Only one study demonstrated that hepatocellular adenoma displays a methylation profile reflective of normal liver tissue [23]. Similarly, we observed that the DNA methylation levels of the four DMMs in benign lesions were low and comparable to those of livers with hepatitis. However, adenomas over 5 cm were reported to have a high risk of transforming into HCC [24].

We found that not all findings obtained by qMSP and MeD-seq were identical, particularly when comparing cirrhotic HCC and cirrhosis. For example, qMSP-based data exhibited similar methylation changes in *CLEC11A* in cirrhotic HCC compared to cirrhosis, while MeD-seq detected distinctive methylation patterns between the two groups. One of the reasons could be the bisulfite conversion process in qMSP, which destroys more than 90% DNA, so smaller input amounts can be used and may affect the results. Also, MeD-seq covers a larger region than qMSP, providing a more comprehensive evaluation of the three DMMs, while the targeted shorter regions of qMSP may present different methylation patterns. This may partially explain the difficulty of reproducibility with DMMs using PCR-related methods in the current research. However, applying sequencing-related techniques to every patient is not feasible, since it is too costly and time-consuming, while PCR-related techniques are cost-effective and can be utilized repeatedly. Therefore, finding specific and sensitive DMMs which could be detected in an accessible and applicable way in clinical practice is important.

Except for *HOXA1*, we partially validated the hypermethylation of *CLEC11A*, *AK055957*, and *TSPYL5* using qMSP or MeD-seq for cirrhotic HCC versus cirrhotic non-HCC livers compared to a previous study [13]. This may be due to tumor heterogeneity, different clinical factors, the choice of study population, and qMSP targets. A number of studies have examined DNA methylation changes between HCC and cirrhotic non-HCC livers in tissues and blood [13,17,25]. However, these studies often detected different DMMs, even with similar sequencing-related techniques [13,25]. It would be interesting to take geographical and ethnical differences into account, since these may have also influenced the study outcomes. It was found that HCC patients from Thailand and France exhibited distinct DNA methylation changes for specific genes in the same study [22], clearly demonstrating the importance of this factor. Interestingly, hypermethylated *HOXA1* exhibited promising results in early-stage HCC detection in cfDNA from blood [16,17]. In this regard, it is important to mention that it is unclear how similar DNA methylation profiles are between blood and liver tissues. We attempted to examine the results of the four DMMs in cfDNA from blood, but were limited by absent or insufficient material.

Although the 146 samples in our study far exceed other studies, small group sizes still hamper sub-group analysis, especially since HCC is highly heterogeneous. Also, in clinical practice, only patients with cirrhosis are identified as high-risk and screened. Acquiring non-cirrhotic HCC tissue is difficult, since patients are often diagnosed at a late stage. Operation is not the first option, but ignoring patients with non-cirrhotic HCC is not good. Therefore, identifying high-risk patients with chronic liver disease without cirrhosis who may develop non-cirrhotic HCC will be crucial in future studies.

## 5. Conclusions

In summary, we found that patients with non-cirrhotic HCC exhibited aberrant DNA methylation levels in liver tissue for *HOXA1*, *CLEC11A*, *AK055957*, and *TSPYL5* as compared to hepatitis, benign lesions, and cirrhosis. Also, gender and cirrhosis status were shown to contribute to the DNA methylation changes of HCC. These novel findings will require further confirmation and refinement in a large clinical study on both genomic DNA from liver tissues and cfDNA from blood.

## Figures and Tables

**Figure 1 cancers-15-04784-f001:**
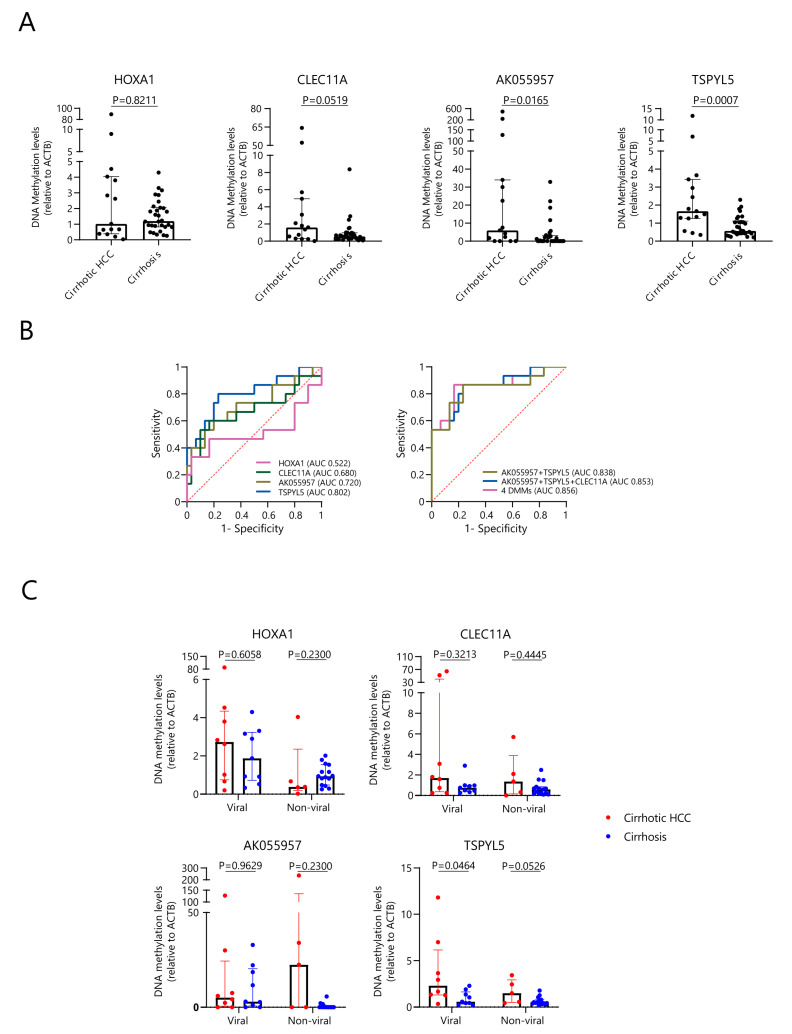
DNA methylation levels of the 4 DMMs in cirrhotic HCC compared to cirrhosis. (**A**) qMSP-based methylation levels, normalized by *ACTB*, are shown for *HOXA1*, *CLEC11A*, *AK055957*, and *TSPYL5* in 15 cirrhotic HCC and 30 cirrhotic non-HCC liver tissues. *AK055957* and *TSPYL5* exhibited higher DNA methylation levels in cirrhotic HCC compared to cirrhosis (*p* < 0.05). (**B**) Regarding the performance of an individual DMM or the combination of DMMs in discriminating cirrhotic HCC from cirrhosis, the panel of 4 DMMs exhibited 87% sensitivity at 80% specificity (AUC 0.86); The red dotted line represents the reference line. (**C**) DNA methylation levels of *HOXA1*, *CLEC11A*, *AK055957*, and *TSPYL5*, normalized by *ACTB*, were measured by qMSP in cirrhotic HCC and cirrhosis tissues with different etiologies, including non-viral (5 HCC, 15 cirrhosis) and viral (8 HCC, 9 cirrhosis). *TSPYL5* showed higher methylation levels in cirrhotic viral HCC compared to cirrhosis (*p* < 0.05).

**Figure 2 cancers-15-04784-f002:**
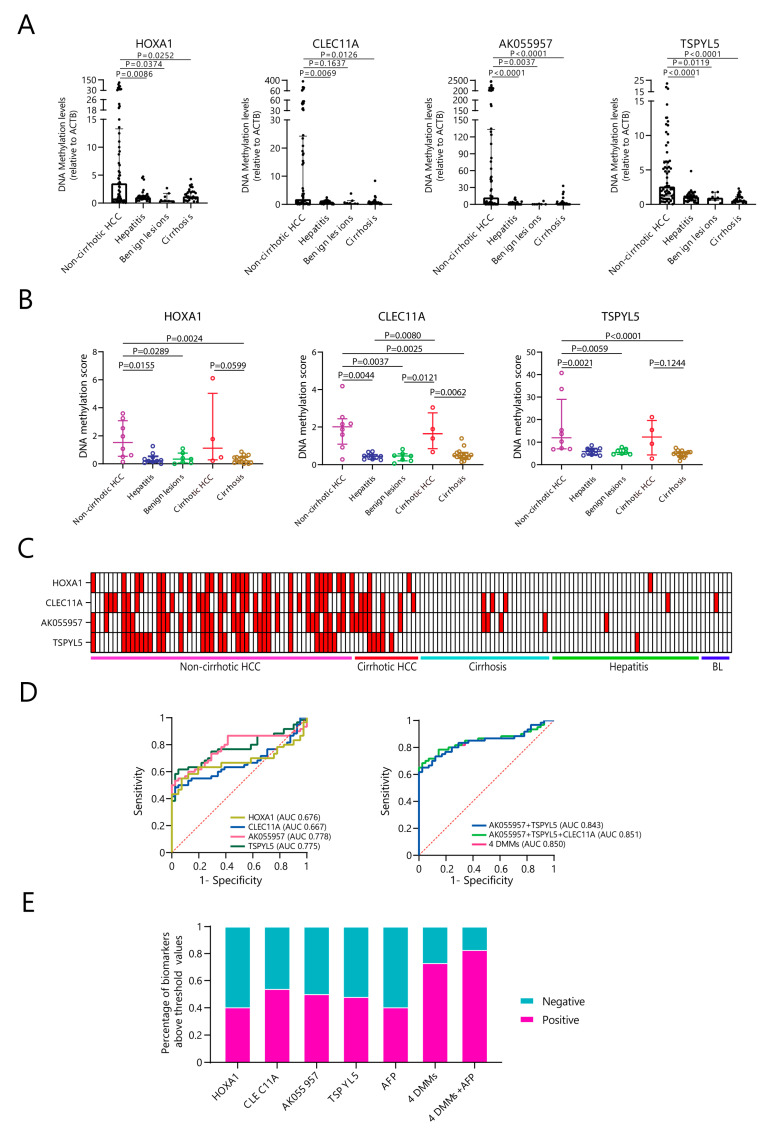
DNA methylation levels of the 4 DMMs in non-cirrhotic HCC, cirrhotic HCC, cirrhosis, hepatitis, and benign lesions. (**A**) DNA methylation levels of *HOXA1*, *CLEC11A*, *AK055957*, and *TSPYL5*, normalized by *ACTB*, were shown by qMSP in 60 non-cirrhotic HCC, 34 hepatitis, 7 benign lesions, and 30 cirrhosis tissues. The 4 DMMs all exhibited higher DNA methylation levels in non-cirrhotic HCC compared to hepatitis and cirrhosis (*p* < 0.05). *HOXA1*, *AK055957*, and *TSPYL5* also showed increased methylation levels in non-cirrhotic HCC compared to benign lesions (*p* < 0.05). (**B**) The DNA methylation scores of *HOXA1*, *CLEC11A*, and *TSPYL5*, calculated by MeD-seq on tissues from alcohol-related liver disease, included 8 non-cirrhotic HCC, 10 hepatitis, 4 cirrhotic HCC, and 15 cirrhosis, as well as 7 benign lesions. All 3 DMMs exhibited hypermethylation scores in non-cirrhotic HCC compared to cirrhosis, hepatitis, and benign lesions (*p* < 0.05). Also, *CLEC11A* showed hypermethylation scores in cirrhotic HCC compared to cirrhosis, hepatitis, and benign lesions (*p* < 0.05). (**C**) The methylation intensity of candidates was confirmed in independent tissue samples; red color indicates methylation above the 95th percentile for the hepatitis group of each DMM (rows) in each tissue sample (columns). White boxes indicate values falling below the 95th percentile in the hepatitis group. (**D**) The performance of individual DMMs and the combination of DMMs in discriminating non-cirrhotic HCC from hepatitis and benign lesions; the panel of 4 DMMs exhibited 78% sensitivity at 80% specificity (AUC 0.85); The red dotted line represents the reference line. (**E**) The percentage of non-cirrhotic HCC patients who had at least one DMM above the 95th percentile in the hepatitis group or AFP higher than 20 ng/µL.

**Table 1 cancers-15-04784-t001:** The performance of DNA methylation markers in terms of discriminating HCC from non-HCC controls.

Biomarkers	Sensitivity, %	Specificity, %	AUC (95%CI)
**Cirrhotic HCC versus cirrhosis**			
HOXA1	47	80	0.522 (0.307–0.738)
CLEC11A	60	80	0.680 (0.492–0.868)
AK055957	60	80	0.720 (0.547–0.893)
TSPYL5	73	80	0.802 (0.657–0.948)
AK055957 + TSPYL5	73	80	0.838 (0.697–0.979)
AK055957 + TSPYL5 + CLEC11A	80	80	0.853 (0.730–0.976)
4 DMMs	87	80	0.856 (0.724–0.987)
**Non-cirrhotic HCC versus controls**			
HOXA1	63	80	0.676 (0.569–0.783)
CLEC11A	55	80	0.667 (0.561–0.772)
AK055957	62	80	0.778 (0.687–0.870)
TSPYL5	63	80	0.775 (0.684–0.866)
AK055957 + TSPYL5	77	80	0.843 (0.766–0.921)
AK055957 + TSPYL5 + CLEC11A	78	80	0.851 (0.775–0.928)
4 DMMs	78	80	0.850 (0.774–0.927)

**Table 2 cancers-15-04784-t002:** qMSP-based DNA methylation percentage above the 95% percentile of hepatitis in non-cirrhotic and cirrhotic HCC tissues compared to other groups.

Disease (N)	HOXA1, %	CLEC11A, %	AK055957, %	TSPYL5, %
**Non-cirrhotic HCC** **(n = 60)**	43.3% (*p* = -)	50.0% (*p* = -)	53.3% (*p* = -)	51.7% (*p* = -)
Cirrhotic HCC (n = 15)	13.3% **(*p* = 0.039)**	46.7% (*p* = 0.816)	40.0% (*p* = 0.815)	33.3% (*p* = 0.567)
Cirrhosis (n = 30)	0 **(*p* < 0.001)**	10.0% **(*p* = 0.002)**	13.3% **(*p* = 0.005)**	0 **(*p* < 0.001)**
Hepatitis (n = 34)	2.9% **(*p* < 0.001)**	2.9% **(*p* < 0.001)**	2.9% **(*p* < 0.001)**	2.9% **(*p* < 0.001)**
Benign lesions (n = 7)	0 **(*p* = 0.037)**	2.9% (*p* = 0.228)	0 **(*p* = 0.037)**	0 **(*p* = 0.037)**
**Cirrhotic HCC** **(n = 15)**	13.3% (*p* = -)	46.7% (*p* = -)	40.0% (*p* = -)	33.3% (*p* = -)
Cirrhosis (n = 30)	0 (*p* = 0.106)	10.0% **(*p* = 0.009)**	13.3% **(*p* = 0.062)**	0 **(*p* = 0.002)**
Hepatitis (n = 34)	2.9% (*p* = 0.218)	2.9% **(*p* < 0.001)**	2.9% **(*p* = 0.002)**	2.9% **(*p* = 0.008)**
Benign lesions (n = 7)	0 (*p* = 1.000)	2.9% (*p* = 0.193)	0 (*p* = 0.121)	0 (*p* = 0.135)

Chi-square tests or Fisher’s exact test were used for testing dichotomous variables when appropriate. Abbreviations: qMSP, quantitative methylation-specific PCR; HCC, hepatocellular carcinoma.

**Table 3 cancers-15-04784-t003:** qMSP-based DNA methylation levels in HCC tumor tissues, classified by clinical factors.

Variables, (n)	HOXA1 Median (*p*)	CLEC11A Median (*p*)	AK055957 Median (*p*)	TSPYL5 Median (*p*)
**Sex**				
Female, (32)	3.22 (*p* = -)	0.53 (*p* = -)	12.64 (*p* = -)	1.45 (*p* = -)
Male, (43)	2.83 (*p* = 0.503)	3.08 (*p* = 0.059)	9.00 (*p* = 0.713)	2.95 **(*p* = 0.035)**
**Age**				
<60, (32)	3.32 (*p* = -)	1.38 (*p* = -)	9.45 (*p* = -)	2.10 (*p* = -)
≥60, (43)	2.85 (*p* = 0.508)	1.80 (*p* = 0.743)	10.48 (*p* = 0.918)	2.44 (*p* = 0.131)
**Etiology**				
Non-viral, (29)	2.01 (*p* = -)	0.78 (*p* = -)	9.91 (*p* = -)	1.86 (*p* = -)
Viral, (16)	4.16 (*p* = 0.069)	1.70 (*p* = 0.337)	8.24 (*p* = 0.552)	3.30 (*p* = 0.070)
Cryptogenic, (30)	2.88 (*p* = 0.821)	3.37 (*p* = 0.283)	19.13 (*p* = 0.753)	2.20 (*p* = 0.992)
**Tumor size** *				
≤5 cm, (22)	3.50 (*p* = -)	1.07 (*p* = -)	3.00 (*p* = -)	2.80 (*p* = -)
>5 cm, (48)	2.73 (*p* = 0.292)	2.66 (*p* = 0.814)	23.38 (*p* = 0.533)	2.33 (*p* = 0.117)
**Cirrhosis**				
Cirrhotic, (15)	1.02 (*p* = -)	1.60 (*p* = -)	5.98 (*p* = -)	1.66 (*p* = -)
Non-cirrhotic, (60)	3.55 **(*p* = 0.025)**	1.87 (*p* = 0.335)	12.19 (*p* = 0.401)	2.59 (*p* = 0.123)

* Five patients without information on tumor size. Statistical differences between different categories were determined by multivariate linear regression; all values were adjusted for each other. Abbreviations: qMSP, quantitative methylation-specific PCR; HCC, hepatocellular carcinoma; HOXA1, homeobox A; CLEC11A, C-type lectin domain containing 11A; TSPYL5, Testis-Specific Y-encoded-Like Protein 5.

## Data Availability

The datasets used and/or analyzed during the current study are available from the corresponding author upon reasonable request.

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
