# Peer review of "Hypermethylation of DNA Methylation Markers in Non-Cirrhotic Hepatocellular Carcinoma"

_cancers, 2023, doi:10.3390/cancers15194784_

Round 1

Reviewer 1 Report

In this paper, FU et al. aim to distinguish non-cirrhotic hepatocellular carcinoma using DNA methylation levels as markers. They analyzed liver tissue samples from 146 patients to investigate the methylation markers of four genes: HOXA1, CLEC11A, AK055957, and TSPYL5. While their sample size is substantial, I find their data to be insufficient to support the conclusions drawn in the paper.

Main Concerns:

The authors attempted to use four reported methylation markers that are inherently random in nature. Furthermore, based on the results in Figure 1, the variations are not very pronounced. Given the considerable number of patient samples, the authors should have employed RNA-seq to identify markers with more pronounced changes. Subsequently, they could have validated these markers at the gene and protein levels, rather than arbitrarily selecting four from the existing literature.

The authors only assessed changes in methylation levels of these four genes themselves, without examining whether there are differences in their expression levels.

Are there better detection methods available, such as directly analyzing changes in certain markers in patient blood for early detection, rather than using liver tissue samples?

The quality of the author's data is subpar. I recommend conducting large-scale screening to identify distinct markers. Following this, patient liver tissue samples should be utilized to detect true markers at the RNA and proteomic levels.

Reviewer 2 Report

The authors have found that patients with non-cirrhotic HCC exhibited aberrant DNA methylation levels in liver tissue for HOXA1, CLEC11A, AK055957, and TSPYL5 as compared to non-HCC tumors. Also, gender and cirrhosis status contribute to the DNA methylation changes of HCC. Limitation of the study: These novel findings will require further confirmation and refinement in a large clinical study both in genomic DNA from liver tissues and cfDNA from blood. This study was executed very well. the authors should discuss the variance of their study to recently published article from European journal of cancer, Fu et al., 2023

Reviewer 3 Report

Fu S et al. assessed the DNA methylation change by quantitative methylation specific PCR (qMSP) and MeD-seq using 4 DNA methylation markers (DMMs; HOXA1, CLEC11A, AK055957, and TSPYL5) in 146 liver tissues samples, and demonstrated a significant DNA methylation change in non-cirrhotic HCC compared to cirrhosis, hepatitis and benign legions. The experimental strategies are well designed and their data are interest. However, several parts are hard to understand in the present version. For example, why the authors described the clinical importance of DNA methylation in cirrhotic cancer (but not non-cirrhotic cancer) in Abstract (comment 4)? I think that the manuscript should be improved.

Comments:

1)   Introduction (page 2, 3rd paragraph) and Results sections (page 4). The authors should explain the reason why 4 DMMs are selected for DNA methylation analysis of tissues samples in this study. It is noted that reference 13 and 15 indicated 10 and 3 DMMs, respectively, both of which are variable markers for analysis of blood samples (cf-DNA).

2)   Page 7, “Except for CLEC11A, increased methylation levels of the other 3 DMMs were also observed in non-cirrhotic HCC compared to benign lesions (p<0.05)”. It looks like very similar bar graphs patterns of hepatitis, benign and cirrhosis of CLEC11A methylation in Figure 2A, but hepatitis and cirrhosis had significant difference against non-cirrhotic HCC. To avoid the confusion (miscalculation etc.), the authors should add the p-value of non-cirrhotic HCC vs benign lesions of CLEC11A, even though this is p>0.05.

3)   What does “non-HCC tumors” mean in the text? Is it a typo as “non-HCC liver tissues”? 

4)   Abstract section, “By PCR or sequencing, we found -----. 87% sensitivity at 80% specificity with an AUC of 0.86, which distinguishes cirrhotic HCC from cirrhosis”. I could not understand Why the authors added these sentences of “cirrhotic HCC” in Abstract? Title and text show the importance of “non-cirrhotic HCC”, leading confusion what the authors want to strengthen their data in Abstract.

5)   I can understand aberrant DNA methylation of 4 DMMs are specific to non-cirrhotic HCC compared to non HCC liver tissues. However, it is uncertain whether these 4 DMMs can distinguish between non-cirrhotic HCC and cirrhotic HCC, because only HOXA1 methylation was specific to non-cirrhotic HCC (Table 2). I read this manuscript many times, but does not reach this answer. Please provide this point.

Round 2

Reviewer 1 Report

The author answered my question, and I have no further inquiries.

Reviewer 3 Report

The authors have addressed the issues I mentioned in the review.